# How Does CSR Activity Affect Sustainable Growth and Value of Corporations? Evidence from Korea

**Kyungtag Lee [1] and Hyunchul Lee [2],***

1   School of Business, Yeungnam University, 280 Daehakro, Gyeongsangbukdo, Gyeongsansi 38541, Korea; marketing@ynu.ac.kr
2   Division of Business Administration, Chosun University, 309 Pilmundaero, Donggu, Gwangju 61452, Korea
*   Correspondence: chul72@chosun.ac.kr; Tel.: +82-62-230-6835

**Abstract:** This study explores the relationship between Korean listed companies' corporation social responsibility (CSR) activities and their sustainable growth and valuation, focusing specifically on the nonlinear aspect. The nonlinear quantile regressions used in this study reported that CSR activities increased corporation value exclusively in the middle-range groups (i.e., $\tau\_25$, $\tau\_50$, $\tau\_75$) of Tobin's q, a proxy for corporation growth and value. However, the linear ordinary least squares (OLS) regression did not indicate similar results. Our findings also showed that CSR activities affect the valuation of Korean listed corporations in a nonlinear, rather than in a linear way. Considering that most prior studies are devoted to reporting linear results from classical ordinary least squares estimations between CSR activities and corporation value, our study fills the gap in the literature. The findings of this study may provide corporation managers and researchers with valuable data concerning a corporation's optimal investment point for their CSR activities for sustainable growth and the maximization of corporation value.

**Keywords:** CSR; Tobin's q; corporation value; corporation growth; sustainability; nonlinearity

## 1. Introduction

Many companies are paying greater attention to corporation social responsibility (CSR) to obtain a sustainable and strategic advantage in very competitive market conditions. CSR is defined as a term of reference and a countermeasure to various issues beyond corporations' economic, technical, and legal requirements. It enables corporations to accomplish not only traditional economic incomes but also social outcomes [1]. However, the definition of corporation business has also expanded as the scope of CSR and social environments change. The aim of CSR is to reduce harms like social costs or generate public goods [2]. In a similar vein, the European Commission report of 2010 defined CSR as a concept that reflects voluntary management of social and environmental issues and positive interaction with interested parties.

Accordingly, the recognition of CSR activities by corporations has recently become more progressive. The international standard for CSR enacted by ISO (International Organization for Standardization) in 2010 is an example of how corporations and society perceive it. Due to the enhanced externalities of corporation activities, which have negative effects on the external environment and interested parity, CSR has played a critical role in corporations' sustainable competitive advantage and growth in the long run [3–5]. Therefore, both academia and the business world have shown considerable interest in CSR, as corporations are required to heed the concerns and social needs of various interested parties [6]. Corporations are faced with urgent requests from advocacy and lobby groups concerning a variety of social, environmental, and political issues. Ultimately, CSR is a much-needed asset for all types of corporations, from small firms to large global businesses [7].

Due to the changing perception of CSR, it has been a crucial theme in management and corporation finance concerning sustainable growth and the valuation of corporations. Prior studies have examined the relationship between CSR activities and corporation performance. Initial studies on CSR focused primarily on the CSR concept and on methods to measure it [8–10]. Since then, several works have addressed the various effects of CSR on corporation management performance [11–14]. Particularly, prior studies on the relationship between corporations' CSR activities and financial performance have attracted considerable attention. In a society in which concepts such as human rights and environmental awareness are essential to business activities, interests concerning the relationship between CSR activities and financial performance is natural [15]. This is an issue that is still being hotly debated in business and finance.

Since Friedman [16] suggested that a corporation's CSR activities have a significant effect on its financial performance, there has been a continuous flow of related studies over the past 40 years. However, these studies reported mixed results [17]. Firstly, some literature addressed the mixed effects of corporations' CSR activities on their financial performance. For example, Barnett and Saloman [18] presented evidence that the direction of the financial performance may vary according to how well corporations combine CSR with their normal business practices, suggesting both positive and negative effects on financial performance. Additionally, the authors indicated a nonlinear relationship between CSR and the financial performance of corporations, concerning the concept of stakeholder influence capacity incorporated in a variety of CSR activities.

Suggesting that CSR activities have a negative effect on financial performance, some studies focused on the fact that the CSR activities of corporations entail increased expenditure. Therefore, corporations implementing CSR are forced to accept a financial loss, compared with rivals that are not involved in CSR activities. According to the trade-off theory, CSR activities force corporations to spend unnecessary money to achieve their CSR performance targets, resulting in a decrease in profitability [19]. This view is supported by several works. For example, Moore [20] found that CSR activities have a negative effect on the financial performance of corporations in the United Kingdom supermarket industry. Lopez et al. [21] also showed that CSR is negatively associated with the financial performance of corporations by using the Dow Jones Sustainability Index as a proxy for CSR performance. These studies argue that corporations need to be more devoted to the maximization of their current value, as pursuing social justification through CSR activities is a behavior that leads only to high expenditures, infringing on the income of shareholders [22]. According to Krüger [23], corporations' CSR activities indicate an agent problem arising in the corporation setting. This means that managers seek to enhance their private interests by implementing CSR activities as a route to obtaining a good reputation with interested parties (e.g., local society, political players, non-governmental organizations, or labor unions) at the expense of shareholder wealth. As a result, corporations' CSR activities have a negative effect on the wealth of shareholders from the perspective of agency costs.

Contrastingly, a number of studies have reported a positive effect of CSR activity on the financial performance of corporations. For instance, Waddock and Graves [12] indicated that corporations may post better financial performances if the interested parties perceive their progressive CSR activities positively. Based on social impact theory, fulfilling the expectations of various interested parties will eventually affect the financial performance of corporations positively [19]. Concerning the market valuation of corporations and CSR disclosures, existing studies such as References [24,25] provide evidence that the disclosure of CSR activities creates a positive excess in stock returns around the disclosing date. Corporations implementing a high level of CSR may enjoy various benefits, such as low-cost financing—a low debt cost due to creditors' favorable response—which could lead to an improvement in their financial performance [26]. A number of works report that the CSR activities of corporations have a positive effect on the value of the corporation, contributing to an increase in wealth for the shareholders since their CSR activities have a positive net present value [27,28].

Most previous studies that present mixed effects of CSR activities on corporation value rely on linear ordinary least squares (OLS) analyses, which does not effectively explain the nonlinearities between corporations' CSR activities and their value. Therefore, this study aims to systematically explore the various effects of CSR activities on sustainable growth and the valuation of Korean listed corporations by applying a nonlinear approach, that is, conditional nonlinear quantile regressions with Tobin's q. The Tobin's q is commonly used as the most representative proxy for corporation growth and value in business finance. The novel estimation technique of nonlinear quantile regressions devised by Reference [29] provides researchers with more elaborate results than those under a classical linear OLS regression. Nonlinearities (heterogeneity) between CSR and corporation value may provide managers with valuable information, enabling them to decide on an optimal investment point for CSR activities to maximize corporation value. The nonlinear nature of our study implies that investors heterogeneously price the discount factor of corporations' CSR activities to different extents of corporation value when they diversify their portfolios. To the best of our knowledge, this study is the first to systematically capture a nonlinear relationship between corporations' CSR activities and their value by using the nonlinear quantile regression technique.

The rest of this study is structured as follows: Section 2 presents the methodology for the study and Section 3 explains the data issues. Section 4 discusses the findings of our empirical analysis and Section 5 concludes.

## 2. Materials and Methods

### 2.1. Conditional Nonlinear Quantile Regression

First, we briefly explain the conditional nonlinear quantile regression of Koenker and Bassett [29] and how it will be utilized to analyze the various effects that Korean listed corporations' CSR activities have on their value.

The conditional nonlinear quantile regression technique provides researchers with a means to find robust estimates, even when the data include serious outliers with a large heterogeneity. This benefit is due to its effective calibration of the entire distribution of the dependent variables that are reliant on the independent variables. Specifically, for sample data with extremely serious outliers that include non-normal disturbances, applying conditional mean estimators to the specified regression model might cause biased estimates. These estimators generally depart from normality. Consequently, the OLS estimates may be substantially biased and inefficient. The quantile regression is more robust to fulfill the normality condition of the sample distribution, as it applies conditional median estimators [30,31].

Fattouh et al. [31] effectively described the estimation procedure of the quantile regression and the major properties of the estimator. For any real random variable $X$, $X$'s distribution function defines the variable as follows:

$$F(x) = \Pr(X \leq x). \tag{1}$$

Here, the $\tau^{th}$ quantile for $0 < \tau < 1$ is defined as

$$Q(\tau) = \inf\{x : F(X) \geq \tau\} \tag{2}$$

Meaning that $(y_i, x_i), i = 1, 2, 3, \ldots, n$, is a sample where $y_i$ is a dependent variable and $x_i$ is a vector of covariates (i.e., regressors). Assuming that the $\tau^{th}$ quantile of the conditional distribution of $y_i$ is linear in $x_i$, the conditional quantile regression model can be specified as follows:

$$y_i = x_i'\beta_\tau + u_{\tau i} \tag{3}$$

$$Quant_\tau(y_i|x_i) \equiv \inf\{y : F_i(y|x)\tau\} = x_i'\beta_\tau \tag{4}$$

$$Quant_\tau(u_{\tau i}|x_i) = 0 \tag{5}$$

where $Quant_\tau(y_i|x_i)$ denotes the $\tau^{th}$ conditional quantile of $y_i$ on the regressor vector $x_i$, and $\beta_\tau$ represents the unknown vectors of coefficients estimated at different values (quantiles) of $\tau \in (0,1)$. $u_\tau$ refers to the error terms at each quantile with a continuously differentiable cumulative distribution function (CDF) and a density function $f_{u\tau}(.|x)$. $F_i(.|x)$ denotes the conditional distribution function of $y$. Changing the value of $\tau$ from 0 to 1, one could estimate the whole distribution of $y$ conditional on $x$. Next, the quantile regression estimator for $\beta_\tau$ can be obtained when the minimization problem below is solved:

$$\min \sum_i^n \rho_\tau(y_i - x_i'\beta_\tau) \tag{6}$$

where

$$\rho_\tau(u) = \begin{cases} \tau u & if, u \geq 0 \\ (\tau - 1)u & if, u < 0 \end{cases} \tag{7}$$

To solve the minimization problem, many studies using this estimation method apply the linear programming technique [29–32]. To determine the standard errors for the regression coefficients at each of the quantiles, the bootstrap method is commonly used. There are two alternative approaches that are generally considered for the bootstrap technique: The design matrix bootstrap technique and the error bootstrap technique [33]. Buchinsky [33] stated that the two approaches arise from quite different assumptions, concerning the form of the asymptotic covariance matrix of $\beta_\tau$. The design matrix bootstrap technique produces a consistent estimator of the asymptotic matrix with no restrictions. However, the error bootstrap technique produces a consistent estimator only under a strict condition of independence [33,34]. Our study's perspective allows us to follow the former approach.

### 2.2. Empirical Specification and Measurement of Variables

This subsection discusses the empirical specifications of our conditional quantile regression analysis, a measurement of variables that enables us to examine the various effects that corporations' CSR activities have on the sustainable growth and the valuation of Korean listed corporations.

### 2.2.1. Empirical Specification of Conditional Quantile Regression

Our conditional quantile regression to analyze the relationship between Korean listed corporations' CSR activity and their value is specified as follows:

$$
\begin{aligned}
Quant_\tau&(Tobin'sQ|X_{it}) \\
&= \alpha_{\tau 0} + \beta_{\tau 1}lncsr_{i,t-1} + \beta_{\tau 2}lnlev_{i,t-1} + \beta_{\tau 3}lnemp_{i,t-1} \\
&+ \beta_{\tau 4}sg_{i,t-1} + \beta_{\tau 6}turnover_{i,t-1} + \sum_{g=1}^{14} group\_dummy \\
&+ \sum_{y=1}^{2} year\_dummy + \epsilon_i
\end{aligned}
\tag{8}
$$

where $Quant_\tau(Y_{i,t} = Tobin'sQ|X_{i,t})$ denotes the $\tau^{th}$ conditional quantile of $Y_{i,t}(= Tobin's\ Q|X_{i,t})$, the value of the corporation in the year $t$. The parameter $\alpha_{0\tau}$ is a constant for each quantile regression. The regressor $lncsr_{i,t-1}$ denotes the first lagged logarithm of the main explanatory variable to proxy the extent of the CSR activities of Korean listed corporations. The regressors $lnlev_{i,t-1}$, $lnemp_{i,t-1}$, and $lnlev_{i,t-1}$ are control variables of the first lagged logarithms of a corporation's leverage and the number of employees. $sg_{i,t-1}$ and $Turnover_{i,t-1}$ are also control variables of the first lag of growth of sales and asset turnover ratio. *group_dummy* and *year_dummy* refer to a group dummy categorized by the Korean industry middle classification system and a time dummy to control for the industrial and time effects in the full sample data, respectively. $\varepsilon_{\tau,i}$ represents the error terms for each quantile.

### 2.2.2. Measurement of Variables

*A. Dependent Variable*

In our quantile regression models, we use Tobin's q, which is a proxy for sustainable corporation growth and value, as the dependent variable. In theory, Tobin's q is measured by

$$= \frac{market\ value\ of\ assets}{by\ substitute\ value\ of\ assets\ at\ the\ end\ of\ the\ fiscal\ year} \qquad (9)$$

However, as an exact measurement of the substitute value of total assets is very unrealistic, we follow the measurement of $q_{it}$ as suggested by References [35–38]. Therefore, the Tobin's q value used in this study is measured by the market value (i.e., book value of liabilities + market value of equity capital) of assets at the end of year (*t*) divided by the book value of assets (i.e., book value of liabilities + book value of equity capital). This measurement process is expressed in Equation (10).

$$= \frac{market\ value\ of\ assets\ (i.e.,\ book\ value\ of\ liabilities\ +\ market\ value\ of\ equity\ capital)}{book\ value\ of\ assets\ (i.e.,\ book\ value\ of\ liabilities\ +\ book\ value\ of\ equity\ capital)\ at\ the\ end\ of\ the\ year} \qquad (10)$$

Chung and Pruitt [39] asserted that using this measurement to determine the approximate value for Tobin's q (i.e., market corporation value), nearly perfectly accounts for the volatility from Tobin's q that Lindberger and Ross [40] suggested.

*B. Exogenous Explanatory Variable*

As the proxy for CSR activities of Korean listed corporations for this study, we used the Best Corporation Citizen Index provided by KEJI (Korea Economic Justice Institute), whose prior studies use Reference [41]. Since 1991, KEJI has been declaring the top 200 Korean corporations by evaluating a variety of aspects and characteristics.

This study uses the revised KEJI index that consists of the soundness of shareholder composition, investment and financing (25 points), fairness (20 points), social contribution (15 points), consumer protection (15 points), environmental management (10 points), and employee satisfaction (15 points) since 2010. The definition of CSR is both complex and complicated [2]. Therefore, KEJI is a very useful indicator for measuring CSR, because CSR is based on various perspectives. Refer to the work of Reference [42] for more details. Our study expects varied effects of CSR on corporation value, which is proxied by Tobin's q. These effects may include no effect or a weak effect on firms with a bigger corporation value, due to greater expenditures for conducting CSR activities, but a positive effect on firms with a smaller corporation value.

*C. Control Variables*

To control for characteristics that may influence corporation value, our regression models include various control variables such as leverage ratio, corporation size, the growth of sales, and turnover, which are characteristics commonly used in corporation finance.

Leverage is measured by the logarithm of the total liabilities divided by the total assets at the end of year $t - 1$. The liabilities variable includes the negative effect of an increase in risk and underinvestment, but simultaneously includes the positive effect of a reduction in corporation taxes and the costs of manager monitoring [36]. Several studies focusing on Korean listed corporations [36,43] discovered that leverage is positively associated with the Korean listed corporations' value. Rajan and Zingales [44] also reported a positive relationship between corporation value and leverage in G7 countries. We use a logarithm of the number of employees divided by the average employee numbers of the corporations at the end of the previous year ($t - 1$) as a proxy of corporation size. Giunta and Trivieri [45] use this variable for proxying corporation size in their corporation financial studies. The growth of sales (*GS*) is used to control for the distinctive effect of leverage on growth rates. Additionally, the effect of corporation growth on the corporation value is calculated by subtracting

one from the sales in year $t − 1$ divided by the sales in year $t − 2$ [36,46]. The asset turnover ratio (Turnover) to control for the efficient use of assets is calculated by sales divided by total assets in year $t − 1$. Lastly, we include group and year dummies in the regression models to account for industry and time effects.

## 3. Data Issues

To empirically examine the relationship between CSR activities of Korean corporations and their value, we focus on corporations listed in the two major Korean securities markets: KOSPI (Korean Stock Price Index) and KOSDAQ (Korean Securities Dealers Automated Quotations) for the full sample period from 1 January 2014 to 31 December 2015. The sample data of Korean corporations listed in the two major markets were acquired from Datastream International.

For accuracy, we filter the listed corporations collected from Datastream International according to the following criteria and discard the corporations that do not qualify:

- Any corporation without a CSR score for the full sample period
- Any corporation for which information is not available in the Datastream International dataset for the full sample periods for measuring related variables
- Any corporation delisted during the full sample periods

The selection procedures provided a final number of 630 corporations from all the Korean corporations listed in the two securities markets. In summary, the raw data for measuring sample corporations' financial variables—such as Tobin's q, leverage, number of employees, the growth of sales, and turnover—were collected from Datastream International and raw data for the CSR indices were collected from KEJI.

## 4. Empirical Findings

This section provides and discusses the empirical results of the nonlinear quantile regressions for the value of Korean listed corporations on their CSR activities while controlling for a variety of corporation characteristic variables. Table 1 provides descriptive statistics for all the variables. The dependent variable of Tobin's q showed a significant gap between the mean (0.890) and the median (0.707), as well as large values of skewness (9.261) and kurtosis (140.242). Skewness is a measure of the asymmetry of a probability distribution function (PDF) of a real-valued random variable and kurtosis is a measure of its tallness and flatness [47].

**Table 1.** Descriptive statistics.

| Variables | Mean | Std. Dev. | Median | Min | Max | Skewness | Kurtosis | Obs. |
|---|---|---|---|---|---|---|---|---|
| Tobin's q | 0.890 | 0.888 | 0.707 | 0.041 | 16.024 | 9.261 | 140.2422 | 630 |
| LNCSR | 4.115 | 0.053 | 4.115 | 3.849 | 4.550 | 0.460 | 10.728 | 630 |
| LNLeverage | 4.281 | 1.108 | 4.115 | −2.120 | 7.730 | −1.111 | 5.322 | 630 |
| LnEmplyoee | 6.070 | 1.391 | 0.925 | 0.010 | 11.470 | 1.178 | 4.757 | 630 |
| Turnover | 0.984 | 0.567 | 0.925 | 0.010 | 3.980 | 1.338 | 6.583 | 630 |
| GS (Growth of Sales) | 0.022 | 0.567 | 0.020 | −1.520 | 1.950 | 2.253 | 27.031 | 630 |

A logged CSR value (*LNCSR*) also presented a high value of kurtosis (10.728). These values—which indicate non-normality of the main variables—supported our use of the quantile regression for an accurate analysis of the relationship between Korean listed corporations' CSR activities and their value.

Table 2 presents the correlation matrix among the independent variables, with no pairs showing a high coefficient. This suggested that there were no serious multi-collinearities across the values, even when all the independent variables were entered simultaneously in our regression models.

Table 3 reports the results of the nonlinear quantile regressions along with the classical OLS regression for comparison. First, the OLS regression in Table 3 estimates a statistically insignificant

coefficient for the *LNCSR* variable of corporations' CRS activities—a main explanatory variable in this study—which suggested no effect of CSR activities on Korean listed corporations' value, proxied by Tobin's q. This unexpected result conflicts with many prior studies that reported a significant positive or negative effect of CSR activities on corporation value (References [12,19,25] for a positive effect and References [19,21] for a negative effect) [20]. Concerning the financial characteristic variables of the corporations in Table 3, the leverage ratio (*LNLeverage*) has a significant negative relationship with the value of the Korean listed corporations (Tobin's q), efficient use of assets (Turnover), and growth of sales (GS).

**Table 2.** Correlation matrix among the exogenous independent variables.

| Variable | LNCSR | LNLeverage | LnEmplyoee | Turnover | SG (Growh of Sales) |
|---|---|---|---|---|---|
| LNCSR | 1 | | | | |
| LNLeverage | −0.200 | 1 | | | |
| LnEmplyoee | 0.214 | 0.302 | 1 | | |
| Turnover | 0.136 | 0.041 | 0.089 | 1 | |
| GS (Growth of Sales) | 0.104 | 0.104 | −0.070 | 0.105 | 1 |

**Table 3.** Quantile regression results for Tobin's q.

| (Obs. = 629) | OLS | Quantile Regressions | | | | | | |
|---|---|---|---|---|---|---|---|---|
| *Variables* | | $\tau_5$ | $\tau_{10}$ | $\tau_{25}$ | $\tau_{50}$ | $\tau_{75}$ | $\tau_{90}$ | $\tau_{95}$ |
| *Constant* | −1.889 (4.019) | 2.786 *** (0.730) | 2.475 *** (0.916) | 0.365 (0.403) | 1.168 (0.921) | −0.477 (1.933) | 0.705 (3.665) | −7.663 (9.479) |
| *LNCSR* | 0.623 (0.953) | 0.252 (0.183) | 0.310 (0.207) | 0.829 *** (0.102) | 0.874 *** (0.230) | 1.057 ** (0.491) | 0.750 (0.954) | 3.001 (2.295) |
| *LNLeverage* | −0.094 ** (0.046) | 0.023 (0.025) | 0.033 (0.022) | 0.021 (0.018) | 0.032 ** (0.015) | −0.038 (0.026) | −0.074 (0.049) | −0.211 * (0.019) |
| *LnEmplyoee* | 0.053 (0.037) | 0.003 (0.014) | 0.005 * (0.012) | 0.012 * (0.006) | 0.014 (0.010) | 0.027 (0.017) | 0.051 (0.054) | −0.096 (0.168) |
| *Turnover* | 0.147 ** (0.074) | −0.009 (0.038) | −0.004 *** (0.029) | −0.002 (0.028) | −0.009 (0.056) | −0.107 * (0.059) | 0.209 * (0.128) | 1.153 *** (0.429) |
| *GS* | 0.500 *** (0.128) | 0.073 (0.109) | 0.09 8(0.101) | 0.278 ** (0.064) | 0.319 *** (0.103) | 0.340 ** (0.165) | 0.374 (0.330) | 0.309 (0.409) |
| *Dummy_Year* | Inclusive | Inclusive | Inclusive | Inclusive | Inclusive | Inclusive | Inclusive | Inclusive |
| *Dummy_Industry* | Inclusive | Inclusive | Inclusive | Inclusive | Inclusive | Inclusive | Inclusive | Inclusive |
| $R^2$ | 0.103 | 0.183 | 0.139 | 0.122 | 0.114 | 0.118 | 0.132 | 0.174 |

*Notes.* The figures in parentheses for the ordinary least squares (OLS) estimators are robust standard errors, while those in parentheses for the quantile regressions are bootstrapped standard errors obtained using 20 replications. ***, **, * indicate significance at the 1%, 5%, 10% levels, respectively.

Compared with the results of no relationship between CSR activities and Korean listed corporation value from the OLS regression, the empirical results from the nonlinear quantile regressions for the whole quantile of Tobin's q in Table 3 showed a completely different picture. Specifically, no significant coefficients were estimated extremely low ($\tau\_5$ and $\tau\_10$) or very high ($\tau\_90$ and $\tau\_95$) by the quantile regressions estimate, which is very different from the classical OLS regression. These results indicated that for corporations with very low or very high coefficients, there was no significant relationship between their CSR activities and their value. For corporations with very low value, CSR activities—for which the companies are forced to incur considerable expenditures—may not be crucial for enhancing their value, since it did not directly contribute to an increase in sales. Additionally, for corporations already in a very high-value group, CSR activities may not be a priority for increasing corporation value, since it did not enhance the public's perception of their efforts in ethics management, which may contribute to a future increase in sales.

However, unlike the OLS regression, the quantile regressions estimated significantly positive values of 0.829, 0.874, and 1.057 for the three middle quantiles $\tau\_25$, $\tau\_50$, and $\tau\_75$, respectively, of the

distribution of the dependent variables of Tobin's q at the 5% and 1% levels. These results suggested that CSR activities had a positive effect on the value of Korean listed corporations in the middle group of Tobin's q. This implied that for mid-valued corporations that are in a growth phase, an investment in CSR activities was essential to increase the corporation value by enhancing public perception of their efforts, thereby contributing substantially to a future sales increase. Our findings concerning the nonlinear effects of CSR activities on corporation value are similar to that of Barnett and Saloman [18], who found that CSR has a nonlinear effect on the financial performance of corporations.

Concerning the effects of the control variables, we observed that growth of sales has a positive effect on the corporation value, which is confirmed by the GS value's significant positive coefficients of 0.278, 0.319, and 0.340 at the 5% and 1% levels for the three middle quantiles of the distribution of Tobin's q, respectively. This result implied that growth of sales through an enhancement of public perception of corporations' efforts in ethics management by implementing CSR activities was crucial for maximizing the value of the corporations in a growth phase. The heterogeneous results across the quantiles of Tobin's q suggested that the nonlinear nature of the effects of the corporations' CSR activities on their value varied according to the corporation value, rather than being similar for all levels of corporation value. This justified our use of the nonlinear quantile regression technique to study the relationship between Korean listed corporations' CSR activities and their value.

This study conducts the *F* test to determine whether the coefficients of various pairs of quantiles are homogenous. To accomplish this, we extended the bootstrap method and constructed a joint distribution, using the bootstrapped standard errors from the same replications (20) as those in the quantile regression estimation in Table 3 (see References [36,48] for a similar application, among others). Table 4 presents the *F* statistics for the coefficients' equality estimated for all the quantile pairs of Tobin's q during the sample period. Without exception, the *F* statistics for all the pairs across the quantiles of the distribution of Tobin's q significantly rejected the null of homogenous coefficients for each pair across all the quantiles at the 1% level. This suggested differentiated effects of the explanatory variable of *LNCSR* (including control variables) on all the pairs. The homogeneity *F* statistics 15.43 on the explanatory variable across all the quantiles also very strongly rejected the null of homogenous coefficients across the quantiles at the 1% level. This suggested differentiated effects of the variables across all the quantiles.

**Table 4.** *F* Statistics for coefficients on the exogenous variables across the conditional quantile regressions.

| | $\tau_5$ | $\tau_{10}$ | $\tau_{25}$ | $\tau_{50}$ | $\tau_{75}$ | $\tau_{90}$ |
|---|---|---|---|---|---|---|
| $\tau_{10}$ | 12.54 *** (0.000) | | | | | |
| $\tau_{25}$ | 433.34 *** (0.000) | 395.90 *** (0.000) | | | | |
| $\tau_{50}$ | 59.18 *** (0.000) | 44.0 *** (0.000) | 69.76 *** (0.000) | | | |
| $\tau_{75}$ | 309.32 *** (0.000) | 529.90 *** (0.000) | 82.16 *** (0.000) | 59.37 *** (0.000) | | |
| $\tau_{90}$ | 117.82 *** (0.000) | 139.08 *** (0.000) | 226.88 *** (0.000) | 412.3 *** (0.000) | 14.91 *** (0.000) | |
| $\tau_{95}$ | 80.02 *** (0.000) | 48.74 *** (0.000) | 215.90 *** (0.000) | 56.92 *** (0.000) | 36.00 *** (0.000) | 5.70 *** (0.000) |
| Homogeneity *F* statistic on the variables across all the quantiles: 15.43 *** (0.000) | | | | | | |

The figures in parentheses are *p*-values. *** indicates significance at the 1% level.

Figure 1 plots the variations in all the estimates from the linear OLS and the nonlinear quantile regressions during the sample period, which we have already discussed.

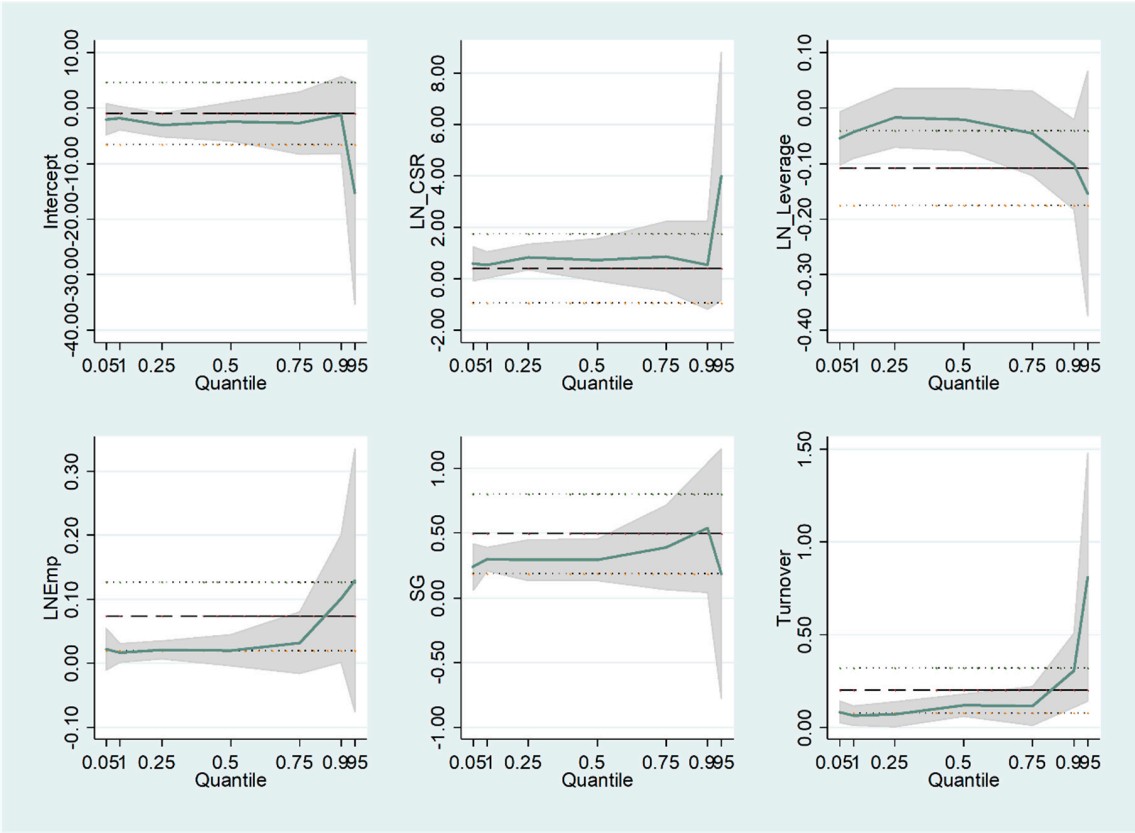

**Figure 1.** Variations in coefficient estimates for the exogenous independent variables. The conditional quantiles (on the x-axis) are the realized correlations distribution conditional on the first lagged exogenous variables, while the intercept (y-axis) ranges from 0 (for the lowest realized correlation) to 1 (for the highest realized correlations). The light gray area around the green lines indicates the 95% confidence intervals (bootstrapped) around the quantiles. The horizontal lines are the OLS estimates with 95% confidence intervals.

## 5. Conclusions

This study aims to explore the relationship between CSR activities and the sustainable corporation value, concentrating on nonlinear relationships. To this end, our study used a conditional nonlinear quantile regression technique that enabled us to analyze the distribution of dependent variables conditional on independent variables. An OLS regression was also run as a comparison.

The nonlinear quantile regressions indicated that the CSR activities of Korean listed corporations contributed to an increase in the corporation value for the middle-range corporation group ($\tau\_25$, $\tau\_50$, $\tau\_75$) exclusively, whereas the linear OLS regression showed a completely different picture. It is worthwhile to note these middle-ranged corporations are generally in a growth phase.

In summary, our findings confirmed that corporations' CSR activities affect the value of Korean listed corporations in a nonlinear (heterogeneous) way rather than in a linear (homogeneous) way, seeing that the CSR activities contributed to the maximization of only middle-ranged corporations. This implies that managers should crucially take into account which phase is their corporations in the corporation development stage when they decide an investment on CSR activities for growth and valuation of their corporation. Thus, our findings provide corporation managers and researchers alike with valuable information for determining corporations' most suitable investment point concerning their CSR activities to promote sustainable growth and maximize corporation value. This study, however, has some limitations. We focus specifically on corporations listed on the Korean securities markets (KOSPI and KOSDAQ). Future studies could investigate the nonlinear relationship between

the CSR activities of corporations and their value by using the quantile regression method with sample corporations listed in advanced markets (e.g., the US, the UK, and Japan). This is beyond the scope of this study, but it could be a suitable avenue for future studies.

**Author Contributions:** Conceptualizing, research methodology, formal analysis, writing; Hyunchul Lee; Conceptualizing, collecting data, data curation, review, project administration; Kyungtag Lee.

**Funding:** This study was supported by the 2016 Yeungnam University Research Grant, grant number 216A580082.

**Acknowledgments:** Kyungtag Lee acknowledges the Yeungnam University which supported the fund for this study.

**Conflicts of Interest:** The author declares no conflict of interest.

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
