# Peer review of "How Does CSR Activity Affect Sustainable Growth and Value of Corporations? Evidence from Korea"

_sustainability, doi:10.3390/su11020508_

Round 1

Reviewer 1 Report

This is a very interesting study and makes a real contribution to the literature. The focus on non-linear relationships is sharp thinking and merits publication and attention. The comments below are aimed at sharpening the conceptual focus of the piece.

The study succeeds or fails based on its ability to identify a clear concept, namely, CSR. Accordingly, the definitional issue is critical to the study. The authors have made a wise choice to use the JEKI.  To validate the choice, they need to explain in the Introduction what the substantive content of CSR is as part of the definition. As they state: "KEJI index that consists of soundness (25 points), fairness (20 points),  social contribution (15 points), consumer protection (15 points), environmental management (10 points), and employee satisfaction (15 points). The widely accepted definition which identifies these as being core to CSR is Sheehy, B. (2014) "Defining CSR: Problems and Solutions" J Bus Ethics.

Further strengthening their argument follows by amending Lines 27 & 32. The definition of CSR has progressed. The definition of CSR has been resolved as "international private business self regulation",  Its aim is"reducing harms like social costs or generating a public good" both in Sheehy, (2014) "Defining CSR: Problems and Solutions" p. 626

Line 39 "Therefore, both academia and the business world have shown considerable interest in CSR," Add reference for interest in CSR by business, academics and government: Sheehy, B (2017) "Conceptual and Institutional Interfaces between CSR, Corporate Law and the Problem of Social Costs," 12 Virginia Law & Business Review 93

Lines 45 -60. Discussion of financial performance and CSR, must reference  Margolis, J. D., Elfenbein, H. A., & Walsh, J. P. (2009). Does it pay to be good—and does it matter? A meta-analysis of the relationship between corporate social and financial performance. Also Soana, M-G (2009) Corporate Financial Performance in the Banking Sector, J Bus Ethic.

The term 'corporates' is inappropriate. It is a colloquial term for 'business organisations'. If the authors wish to focus on listed companies, they are dealing with large industrial organisations. They should also note that there is a separate research stream on SME's e.g. Spence, L Spence, L. J. 1999. Does Size Matter? The State of the Art in Small Business Ethics Business Ethics: A European Review 8(3): 163–174

Author Response

Response to Reviewer 1

Above all else, the authors thank you for your constructive and thoughtful comments. These have helped substantially to improve the exposition of the paper and enhance its overall quality. Below we address each of your comments (in italics) in turn and explain our responses. Our changes in the revised manuscript are highlighted in yellow

Comments and Suggestions for Authors

The study succeeds or fails based on its ability to identify a clear concept, namely, CSR. Accordingly, the definitional issue is critical to the study. The authors have made a wise choice to use the JEKI.  To validate the choice, they need to explain in the Introduction what the substantive content of CSR is as part of the definition. As they state: "KEJI index that consists of soundness (25 points), fairness (20 points),  social contribution (15 points), consumer protection (15 points), environmental management (10 points), and employee satisfaction (15 points). The widely accepted definition which identifies these as being core to CSR is Sheehy, B. (2014) "Defining CSR: Problems and Solutions" J Bus Ethics.

Point taken.

The definition of CSR could be complex. The KEJI index used in this study composes of a variety of activities concerned with CSR. We added the related definitions in this revised version. You can check this at line 209 of p. 6 (reference 2).   

Further strengthening their argument follows by amending Lines 27 & 32. The definition of CSR has progressed. The definition of CSR has been resolved as "international private business self regulation",  Its aim is"reducing harms like social costs or generating a public good" both in Sheehy, (2014) "Defining CSR: Problems and Solutions" p. 626

Good reference.

We added a previous study associated with the aim of the CSR in this revision. You can see line 31 of p.1 (reference 2). 

Line 39 "Therefore, both academia and the business world have shown considerable interest in CSR," Add reference for interest in CSR by business, academics and government: Sheehy, B (2017) "Conceptual and Institutional Interfaces between CSR, Corporate Law and the Problem of Social Costs," 12 Virginia Law & Business Review 93G

Appropriate reference.

We added a new reference related to this issue. Please check the reference 6 at the line 42, p. 1. 

Lines 45 -60. Discussion of financial performance and CSR, must reference  Margolis, J. D., Elfenbein, H. A., & Walsh, J. P. (2009). Does it pay to be good—and does it matter? A meta-analysis of the relationship between corporate social and financial performance. Also Soana, M-G (2009) Corporate Financial Performance in the Banking Sector, J Bus Ethic.

Agreed!

To effectively respond to this issue, we added a new reference at the line 48-50, p. 2 (references 8, 9).

The term 'corporates' is inappropriate. It is a colloquial term for 'business organisations'. If the authors wish to focus on listed companies, they are dealing with large industrial organisations. They should also note that there is a separate research stream on SME's e.g. Spence, L Spence, L. J. 1999. Does Size Matter? The State of the Art in Small Business Ethics Business Ethics: A European Review 8(3): 163–174

Agreed!

Following the other reviewer’s comment for this issue, we replaced the world ‘corporate(s)’ with the word ‘corporation(s) throughout the (revised) paper.

Lastly, needless to say having received the service for the1st round submission, we seriously moderated  the English editing for this revision.

Again, we really appreciate your constructive comments!!

Reviewer 2 Report

Interesting. Thanks for the opportunity to review.

A minor suggestion: you use "corporates" where an American author would likely use the word "corporations" - so I would ask the editors what they prefer.

Also, you seem to imply in the conclusion that South Korea is not a fully developed market, whereas, in my understanding as an outside observer, it seems to be.

Overall, I like this piece and think it should be published after some small relatively small changes. Here are the more significant suggested revisions. As to more significant points:

(1) I would suggest better defining all the terms, including linear and non-linear, and Tobin's Q. Plus, at both the beginning and conclusion, include a few sentences that would be comprehensible to a layperson as to the meaning of your results, and why they matter.

(2) Here are some citations to previous related works you may want to include:

Use of Tobin's Q in the context of studies of CSR (it is indeed rare, but is supported by its adoption and use in other studies related to CSR:

Walsh, C. & Sulkowski, A.J. (2010). A Greener Company Makes for Happier Employees More So Than Does a More Valuable One: A Regression Analysis of Employee Satisfaction, Perceived Environmental Performance and Firm Financial Value. Interdisciplinary Environmental Review, 11 (4), 274-282).

Support for your research agenda, and related findings:

Waddock, S.A. & Graves, S.B. (1997). The Corporate Social Performance-Financial Performance Link. Strategic Management Journal, 18:4, 303-319.

Wu, J., Liu, L. & Sulkowski, A.J. (2011). Environmental Disclosure, Firm Performance, and Firm Characteristics: An Analysis of S&P 100 Firms. Journal of the Academy of Business and Economics, 10 (4), 73-84.

Wei, L., Wenjun, W., Sulkowski, A. J. & Wu, J. (2011). The Relationships between Environmental Management, Firm Value and Other Firm Attributes: Evidence from Chinese Manufacturing Industry. International Journal of Environment and Sustainable Development, 10 (1), 78-95.

(3) Lines 98-100: "I think you want to revise this sentence: "The novel estimation technique of nonlinear quantile regressions was devised by [24] and encourages more ingenious results than those under a classical linear OLS regression."

(4) Lines 121-122 - this phrase could be clearer: "The quantile regression
122 is more robust to departure from normality,"

(5) Roughly lines 130-145: check the justification of the text... it seems to be centered for more of the text than intended.

(6) Line 203: could you explain briefly "soundness"? What does that mean?

(7) Line 252 can you define these terms briefly, please: skewness (9.261) and kurtosis (140.242).

(8) Lines 327-329: why are there 2 Figure 1 labels?

Figure 1.plots the variations in all the estimates from the linear OLS and the nonlinear quantile
regressions during the sample period, which we have already discussed.
Figure 1. Variations in Coefficient Estimates for the Exogenous Independent Variables.

(9) As far as Figure 1 is concerned: could you explain the meaning of the gray shading in the images of the graphs?

(10) Overall, in the final 2 sections, I am still wondering how many readers will understand the (1) findings and (2) their significance... could these be "translated" so that a layperson can understand?

I think that will make the article more valuable and impactful.

Author Response

Response to Reviewer 2

Above all else, the authors thank you for your constructive and thoughtful comments. These have helped substantially to improve the exposition of the paper and enhance its overall quality. Below we address each of your comments (in italics) in turn and explain our responses. Our changes in the revised manuscript are highlighted in yellow.

Comments and Suggestions for Authors

A minor suggestion: you use "corporates" where an American author would likely use the word "corporations" - so I would ask the editors what they prefer.

Would be better.

So, the authors replaced the world ‘corporate(s)’ with the word ‘corporation(s) throughout the (revised) paper.

Also, you seem to imply in the conclusion that South Korea is not a fully developed market, whereas, in my understanding as an outside observer, it seems to be.

Sounds reasonable.

However, it would be worthwhile to note that the major index (KOSPI index) of the Korean securities market is not still listed at the MSCI index, the representative index for the advanced market. So, for a matter of a rigorous study, we view the market of South Korea as one of leading developing markets.  

I would suggest better defining all the terms, including linear and non-linear, and Tobin's Q. Plus, at both the beginning and conclusion, include a few sentences that would be comprehensible to a layperson as to the meaning of your results, and why they matter.

Point taken.

We tried to better define the terms. You review this at the lines 100-103, p. 3 in this revised version.    

(2) Here are some citations to previous related works you may want to include:

Use of Tobin's Q in the context of studies of CSR (it is indeed rare, but is supported by its adoption and use in other studies related to CSR:

Walsh, C. & Sulkowski, A.J. (2010). A Greener Company Makes for Happier Employees More So Than Does a More Valuable One: A Regression Analysis of Employee Satisfaction, Perceived Environmental Performance and Firm Financial Value. Interdisciplinary Environmental Review, 11 (4), 274-282).

Appropriate reference!

We added the appropriate reference in the revision. You can check the new reference at the line 189, p.6.

Support for your research agenda, and related findings:

Waddock, S.A. & Graves, S.B. (1997). The Corporate Social Performance-Financial Performance Link. Strategic Management Journal, 18:4, 303-319.

Incorporated in the original and revised versions.

Wu, J., Liu, L. & Sulkowski, A.J. (2011). Environmental Disclosure, Firm Performance, and Firm Characteristics: An Analysis of S&P 100 Firms. Journal of the Academy of Business and Economics, 10 (4), 73-84.

Appropriate!

We added the reference in this revision. Please see it at the line 89, p. 6 in this revision.  

Wei, L., Wenjun, W., Sulkowski, A. J. & Wu, J. (2011). The Relationships between Environmental Management, Firm Value and Other Firm Attributes: Evidence from Chinese Manufacturing Industry. International Journal of Environment and Sustainable Development, 10 (1), 78-95.

Fine!

We also added the reference in this revision. Please see it at the line 50, p. 2 in this revision.  

(3) Lines 98-100: "I think you want to revise this sentence: "The novel estimation technique of nonlinear quantile regressions was devised by [24] and encourages more ingenious results than those under a classical linear OLS regression."

Good point.

The revision for this are done at the lines 101-103, p. 3 in this revision.

(4) Lines 121-122 - this phrase could be clearer: "The quantile regression
122 is more robust to departure from normality,"

Point taken.

Please review the revision for this at the lines 124-125, p. 3 in this revision.

(5) Roughly lines 130-145: check the justification of the text... it seems to be centered for more of the text than intended.

Sound reasonable.

Please note that the part is for a general and brief explanation for the theoretical background of the conditional quantile regression devised by the developers (indicated in the text). 

(6) Line 203: could you explain briefly "soundness"? What does that mean?

Good point.

We detailed the soundness at the line 205-206 in this revsion. 

(7) Line 252 can you define these terms briefly, please: skewness (9.261) and kurtosis (140.242).

Sounds plausible for laypersons in econometrics.

So, we more explained them in this revision. Please see the lines 255-256 1 at page 6.  

(8) Lines 327-329: why are there 2 Figure 1 labels?

Figure 1.plots the variations in all the estimates from the linear OLS and the nonlinear quantile
regressions during the sample period, which we have already discussed.
Figure 1. Variations in Coefficient Estimates for the Exogenous Independent Variables.

That should be an editing mistake. Corrected in this revised version. So sorry for the absurd mistake! 

(9) As far as Figure 1 is concerned: could you explain the meaning of the gray shading in the images of the graphs?

Good point.

We explained it in the notes below the figure.

(10) Overall, in the final 2 sections, I am still wondering how many readers will understand the (1) findings and (2) their significance... could these be "translated" so that a layperson can understand?

Great point.

The authors tried to make this issue clearer in this revision. Please review the lines 352-354 in the conclusion section, p 11. 

Again, we really appreciate your constructive comments!!
